# Bacterial adaptation is constrained in complex communities

Thomas Scheuerl [1]*, Meirion Hopkins[1], Reuben W. Nowell [1,3], Damian W. Rivett [1,2],
Timothy G. Barraclough [1,3] & Thomas Bell [1]

A major unresolved question is how bacteria living in complex communities respond to environmental changes. In communities, biotic interactions may either facilitate or constrain evolution depending on whether the interactions expand or contract the range of ecological opportunities. A fundamental challenge is to understand how the surrounding biotic community modifies evolutionary trajectories as species adapt to novel environmental conditions. Here we show that community context can dramatically alter evolutionary dynamics using a novel approach that 'cages' individual focal strains within complex communities. We find that evolution of focal bacterial strains depends on properties both of the focal strain and of the surrounding community. In particular, there is a stronger evolutionary response in low-diversity communities, and when the focal species have a larger genome and are initially poorly adapted. We see how community context affects resource usage and detect genetic changes involved in carbon metabolism and inter-specific interaction. The findings demonstrate that adaptation to new environmental conditions should be investigated in the context of interspecific interactions.

[1] Department of Life Sciences, Imperial College London, Silwood Park Campus, Ascot, Berkshire SL5 7PY, UK. [2] Department of Natural Sciences, School of Science and Engineering, Manchester Metropolitan University, Manchester, UK. [3]Present address: Department of Zoology, University of Oxford, 11a Mansfield Road, Oxford OX1 3SZ, UK. *email: tscheuer@ic.ac.uk

Species' niches are constantly adjusting in response to changing environmental conditions[1–6]. In shifting environments, lineages can suddenly find themselves maladapted[7] and experience positive selection to adapt to new conditions[3]. Populations can exploit new ecological opportunities by altering their physiological state[8,9] and by evolving via new beneficial mutations[3,10] or standing genetic variation[11,12]. The intrinsic ability of a population to explore the fitness landscape depends on its mutation rate[13], population size[14,15], the degree of maladaptation[7], and the malleability of its genome[16]. However, the role of intrinsic factors in adaptation could depend on the extrinsic environmental context[17–22]. Ecosystems contain many species that might constrain adaptation by competing for resources that would otherwise be available[4,16,23–25] and force populations to exploit less rewarding niches[16,26,27]. Alternatively, other species might facilitate adaptation by generating new niches[28,29] or suppressing competitors[18]. Observations from nature[4,5,30–32] and laboratory studies using simple artificial communities[2,33–35] indicate that biotic interactions can alter evolutionary responses[16,17,19,20,36]. While these studies have documented the impact of the community on evolutionary dynamics, none have systematically altered both extrinsic factors (across multiple communities) and intrinsic factors (across multiple lineages) to assess their relative roles during adaptation[31,37].

We collected environmental samples from pools of rainwater, which have been used extensively as natural micro-ecosystems for addressing ecological[38] and evolutionary questions[2,33]. Focal bacterial strains isolated from the samples were caged in dialysis bags (see Supplementary Methods, Supplementary Fig. 1) and suspended in laboratory aquatic microcosms containing intact rain-pool bacterial communities for a period of 5 months. The dialysis bags physically separated the focal strains from the surrounding community, while growing on beech leaf medium (Supplementary Fig. 2), preventing direct contact or horizontal gene transfer, but allowing chemical interactions (e.g. resource competition, cell–cell signalling)[39]. This experimental design allowed us to track the adaptation of 22 focal bacterial strains (Supplementary Fig. 3) as they interacted chemically with hundreds of other taxa in eight complex communities. We grew the strains and communities in a boiled-leaf medium modified to have low $pH = 5.5$. The pH is among the most important determinants of microbial community composition across many ecosystems[40] and varies substantially in water-filled tree holes[41], so is likely to be an important selective pressure. The pH of the medium imposed an abiotic environmental stress on the bacteria, since most ancestral isolates and communities grow better at higher pH (Supplementary Fig. 4a). Consequently, strains were under the same abiotic selection pressures to adapt to low pH but experienced a variety of biotic interactions with the different background communities. Rather than regularly transferring a small proportion of the population to fresh media, in the style of classical experimental evolution, we replaced only 10% of the medium once per week to mimic natural conditions. We anticipated that labile carbon resources (e.g. sugars) would therefore be used up over time and that competition for more recalcitrant resources (e.g. cellulose) would intensify. We created a living archive of frozen samples collected from the dialysis bags over the 5 months. By comparing the growth, metabolic phenotype (i.e. degradation of labile vs. recalcitrant carbon sources) and the genotype of the evolved population to those of the ancestral strains, we assessed both the tempo and mode of evolution of focal strains.

We see that focal strains have a higher evolutionary capacity in low diversity communities. The evolutionary response of the strains depended both on the combined impacts of the identity of the focal strain, properties of the strain (e.g. genome size), properties of the background community (e.g. biodiversity), and interactions among these factors. The findings suggest that the background community should be considered when investigating bacterial adaptations to new environments.

## Results and discussion

**Focal strain evolution was different in each community**. We measured performance (competitive fitness) of the evolved populations relative to the ancestral population when both were grown in separate dialysis bags embedded within the community at the end of the experiment (see Supplementary Fig. 1). The focal strains (rows of Fig. 1) had wide variation in performance across the different communities (columns of Fig. 1). Some evolved populations adapted strongly to laboratory conditions and their population densities greatly increased compared to ancestral populations (e.g. *Raoultella* sp.2, *Curtobacterium* sp.1, *Rhizobium* sp.1), while others showed weak or non-significant adaptation (e.g. *Serratia* sp., *Microbacterium* sp.). Similarly, some communities (columns of Fig. 1) were permissive in allowing focal strains to adapt (e.g. communities 1, 3, 4, and 7), while other communities were restrictive (e.g. communities 2, 5, 6, and 8). Similarities in evolvability were not simply explained by genetic similarity: *Raoultella* sp.1 clearly had the capacity to adapt rapidly in some communities, but *Raoultella* sp.2 did not show the same pattern. Instead, the evolvability of most species depended on the interaction between focal strain and community, which explained 38% of the observed evolutionary change, while focal strain and community alone only explained 13% each (Fig. 1, pie plot, Supplementary Table 1).

**Extrinsic and intrinsic properties determine evolution**. We quantified the impact of extrinsic factors on the evolvability of the focal strains by characterising the communities according to their diversity (Shannon's Index) and their ecological 'robustness' in response to an environmental perturbation. Ecological robustness was quantified as the degree to which the instantaneous activity of each ancestral community was reduced by the low pH conditions (see Supplementary Methods, Supplementary Fig. 4); robust communities were little impacted by changes to pH. Our findings demonstrate that focal strains evolved higher performance in communities with low diversity (low Shannon's index, Fig. 2a). The result implies that, while bacterial strains have great capacity to adapt, as has been shown in numerous simple laboratory systems[1–3,7,16,33], that capacity is reduced as communities become more diverse. This does not appear to result simply from the richness (number of taxa) but is a consequence of an uneven distribution of abundance. Several factors could contribute to this pattern. For example, focal species might adapt more easily to a few dominant species (low Shannon's index) than to multiple moderately abundant species. Diversity and community robustness both mainly negatively affected resource degradation (Fig. 2a), so competition for resources might be a factor that limits adaptation of the focal strain in diverse communities[27]. Alternatively, low-diversity communities themselves might adapt less rapidly to new conditions because a greater proportion of the species are at low abundance, giving the focal species a relative advantage. We also observed that evolved strains had a greater capacity to use chitin as a resource in communities with low robustness. This may indicate that chitin was more available in communities that were struggling with pH 5. While we cannot definitively separate these mechanisms using the available data, further experiments could test whether resource competition was driving these patterns, for example by creating environments where resources were not limiting to growth. Regardless of the

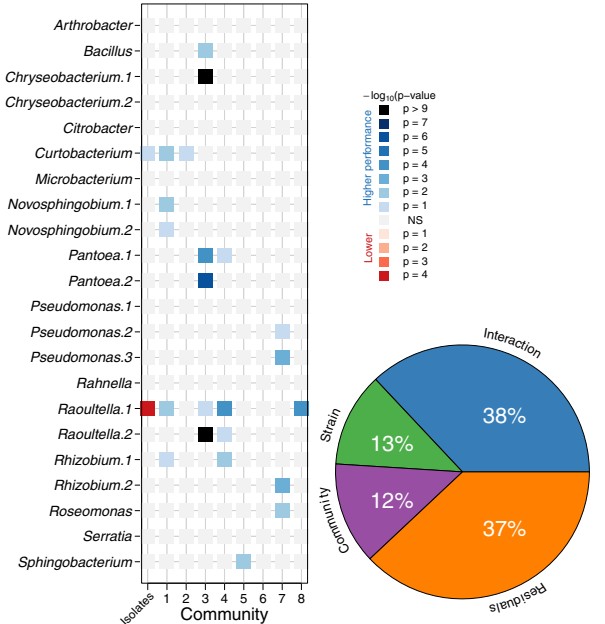

**Fig. 1 Evolution of focal strains in complex communities.** Each row is a focal strain, and each column is the community in which it evolved. Performance was measured as the growth of the evolved focal strains divided by the growth of the ancestral strain in the evolved community over a period of 2 weeks. Blue squares indicate that the evolved focal strain grew significantly better than its ancestral counterpart, while red squares indicate that the evolved strain grew significantly worse. Non-significant differences are not shown. Colour intensity reflects the significance level ($n = 4$). The pie plot shows the overall amount of variance explained by species, communities, and their interaction. Details of the statistical analysis are given in the Supplementary Methods. Source data are provided as a Source Data file.

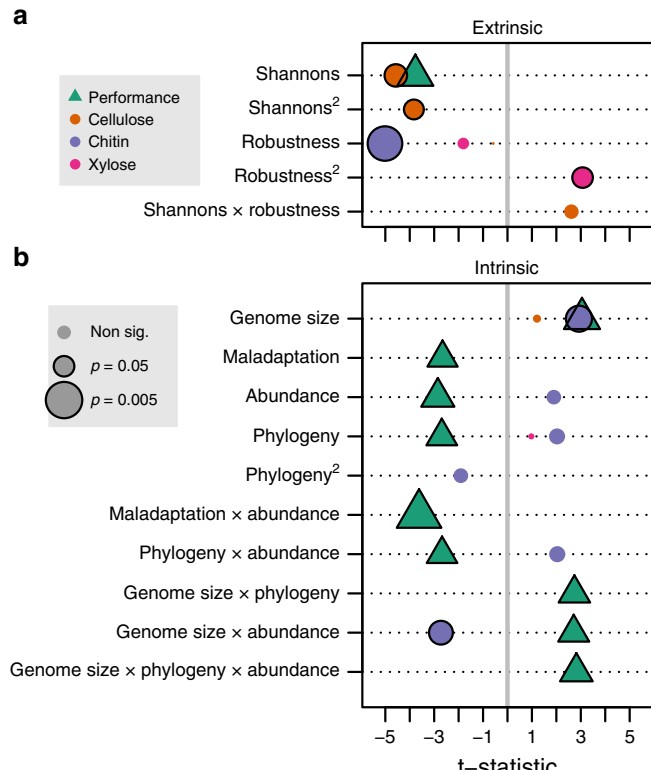

**Fig. 2 Impact of extrinsic and intrinsic factors on performance and enzyme usage. a** We modelled how each response variable (different colours in the figure) was impacted by extrinsic factors and their interaction (rows in the figure), and **b** showing the effect of intrinsic factors and their interaction. The $t$-statistics associated with each factor are shown, with positive/negative $t$-statistics indicating a positive/negative association between the factor and the response variable. Only those factors that were retained in the best model are shown. Non-significant interactions that were not retained for any factor are not shown. A full explanation of the statistical modelling is given in the Supplementary Methods. Non-linear trends are indicated by the square of the variable (e.g. Shannons$^2$); detailed definition of the extrinsic and intrinsic factors (Shannons, Robustness, Phylogeny, etc.) are given in the Supplementary Methods. Source data are provided as a Source Data file.

underlying cause, the result shows that properties of the background community alter focal strain adaptation.

We also quantified four intrinsic properties of the ancestral focal strains: (i) their degree of maladaptation when encountering the low pH conditions in monoculture (i.e. instantaneous reduction in activity at pH 5.5 compared to pH 7, Supplementary Fig. 4b), (ii) their mean abundance across all communities, (iii) their inferred genome size, and (iv) their phylogenetic distance from one of the focal strains (see the "Methods" section). We found that strains with larger genomes were associated with a greater capacity to adapt to the laboratory environment (Fig. 2b). Larger genomes are more likely to contain pre-adaptations to new conditions[42], have higher number of mutations per generation, and might contain multiple gene copies, allowing one copy to change with little fitness cost[42]. Focal strains with the highest evolvability were also relatively rare on average in their native communities and were more closely related phylogenetically to the strain with the highest average evolvability (*Raoultella*), thus potentially occupying underutilised niches. Finally, we found that maladapted focal strains evolved more, which might reflect that well-adapted species experience little selection pressure and hence respond slowly to selection[7].

The extrinsic and intrinsic performance correlates point toward a key role of pre-adaptation in providing the opportunity for further adaptation. Under this model, ecological opportunity is granted by extrinsic factors (e.g. a low-diversity community), while the capacity to adapt is governed by pre-adaptation (degree of maladaptation) and the available toolkit (genome size). Our experimental design is unable to confirm a causal link between

these factors and the focal populations, but we hypothesised that the basis for extrinsic and intrinsic effects revolved around pre-adaptation for resources within the microcosms, and whether the focal strains could capitalise on the resource environment left underexploited by the background community. We characterised the resource niches of the ancestral and evolved focal strains by quantifying their potential to degrade labile (xylose), recalcitrant (cellulose), and intermediate (chitin) substrates (Supplementary Table 2) found in tree-hole environments. We found that the intrinsic and extrinsic factors had significant impacts on the evolution of resource usage, particularly related to cellulose and chitin degradation (Fig. 2). The ability of the focal strains to exploit the more recalcitrant resources was constrained by extrinsic factors. Cellulose degradation was constrained in more diverse communities and chitin was constrained in more robust communities. Improved chitin degradation was also associated with some intrinsic factors, notably genome size. There was little evolution to use xylose (the most labile substrate), perhaps because rates of degradation could not be improved or because xylose was rare late in the experiment when we assume most of the labile substrates had been degraded.

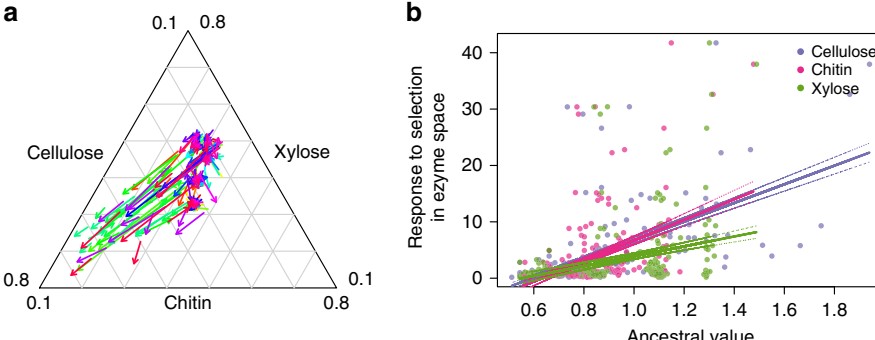

**Fig. 3 Evolution of substrate degradation capacity and influence of ancestral resource utilisation. a** We measured the ability of each focal strain to degrade three substrates (xylose, chitin, cellulose). The base of the arrow indicates the mean ability of the ancestor to degrade the substrates, and the arrowhead indicates the mean ability of the evolved strain to degrade the substrate. Evolved and ancestral substrate degradation was averaged across replicates. Colours correspond to the focal strain. **b** We investigated how ancestral resource utilisation influences the response to selection. The x-axis is the resource utilisation of the ancestral focal strains. The y-axis is the response to selection, measured as the change in resource utilisation from the ancestral to the evolved for each focal strain in every community. These distance values correspond to the length of the arrows in Fig. 3a. Lines represent best fit lines and dotted lines represent 95% confidence intervals highlighting the steeper slope for cellulose and chitin (see Supplementary Table 3 for statistics). Source data are provided as a Source Data file.

**Evolved focal strains used recalcitrant resources**. A more detailed look at how substrate usage evolved revealed a remarkably consistent evolutionary trajectory (Fig. 3a). First, for most strain–community combinations there was little change in enzyme activity, indicating that the strains were either constrained to their ancestral niche by intrinsic or extrinsic factors, that the ancestral niche was a local optimum, or that there was insufficient time for significant evolutionary change (see the "Methods"section). For those focal strains that did respond to selection with a niche shift, there was a consistent change in substrate usage, with an increased ability to use cellulose mainly at the cost of chitin, and with less impact on xylose. Overall, the result is plausible, since the low replacement rate of fresh media would likely have led to the accumulation of recalcitrant substrates[38]. Finally, there was a tendency for the response to selection (length of the arrows in Fig. 3a) to be highest for those ancestral focal strains that started with the highest cellulose and chitin degradation ability (Fig. 3b, Supplementary Table 3). The result again implies an important role of pre-adaptation to the resource environment[43,44].

**Genetic variants underlying adaptation**. We used genome sequencing to determine whether genetic changes underlie the observed phenotypic changes. In particular, we conducted shotgun sequencing of populations of evolved *Raoultella* sp.1 and sp.2 across the communities (Fig. 4, Supplementary Table 4) and compared these to the genomes sequenced from the ancestral populations. This approach revealed several genetic variants, including variants that were gained over the course of the experiment. In addition, several genetic variants arose during the initial growth of the originally clonal ancestral population and were either lost or changed in frequency during the experiment, as has been observed in comparable studies[45,46]. There was evidence that some of the variants were involved in carbon metabolism (*kdgT* gene; a 2-keto-3-deoxy-D-gluconate transporter that regulates pectin metabolism) and in inter-specific interactions (*pipB2* gene, a secreted effector protein that can modify cell interactions), consistent with our observations that substrate degradation and community interactions could explain the evolutionary trajectory of the focal populations. However, most of the genes involved have not been annotated, so further work will be needed to confirm which of the variants are responsible for the changes in focal strain performance and metabolic phenotypes we

observed. There was no clear impact of community diversity on distribution and identity of the genetic variants in evolved *Raoultella* strains, which may be because selection was targeting similar loci in different communities. Alternatively, some of the phenotypic changes we observed that were linked to changes in community diversity could have been due to behavioural (rather than genetic) modifications in the focal strains. Finally, the reduced statistical power to detect correlations in this smaller sample set could have obscured a signal, which could be remedied by more comprehensive sequencing across the focal strains or by exploring alternative statistical models.

**Final communities were dominated by similar taxa**. To see whether the biotic environment changed during the experiment, we determined the change in the background communities by 16S amplicon sequencing. Although all final communities were distinct and highly diverse, initial differences between the communities dissipated during the experiment, and final communities converged such that there was no significant difference among the communities at the conclusion of the experiment (Fig. 4a, PERMANOVA, d.f. = 7, F-value = 0.7679, p = 0.987). In particular, all of the final communities were dominated by *Erwinia* sp., *Klebsiella* sp., *Serratia* sp., *Pseudomonas* sp., or *Pantoea* sp. (Fig. 4b). A likely explanation of these dynamics is ecological sorting of species with environmental filtering[47], but the phenotypic and genetic evolution observed in the focal strains suggest that evolutionary processes in the communities could also play a role. The result also implies that the variation in response to selection of the focal strains in the different communities (Fig. 1) likely resulted from differences in community composition early in the experiment.

Although different abiotic selective pressures would undoubtedly produce different results in terms of which strains and genes evolved, we suggest that these results illustrate general features of bacterial communities: that extrinsic and intrinsic impacts on adaptation cannot be viewed in isolation, since the interaction between these two components is considerably more important than each one alone (Fig. 1). The change in substrate degradation rates by the focal strains (Fig. 3) provides a likely mechanism by which this occurs. Communities where competition is high (diverse communities or communities that contain a high relative abundance of congeneric taxa) act to constrain adaptation, likely through pre-emption of the available niche space[4,27,30,33,48,49].

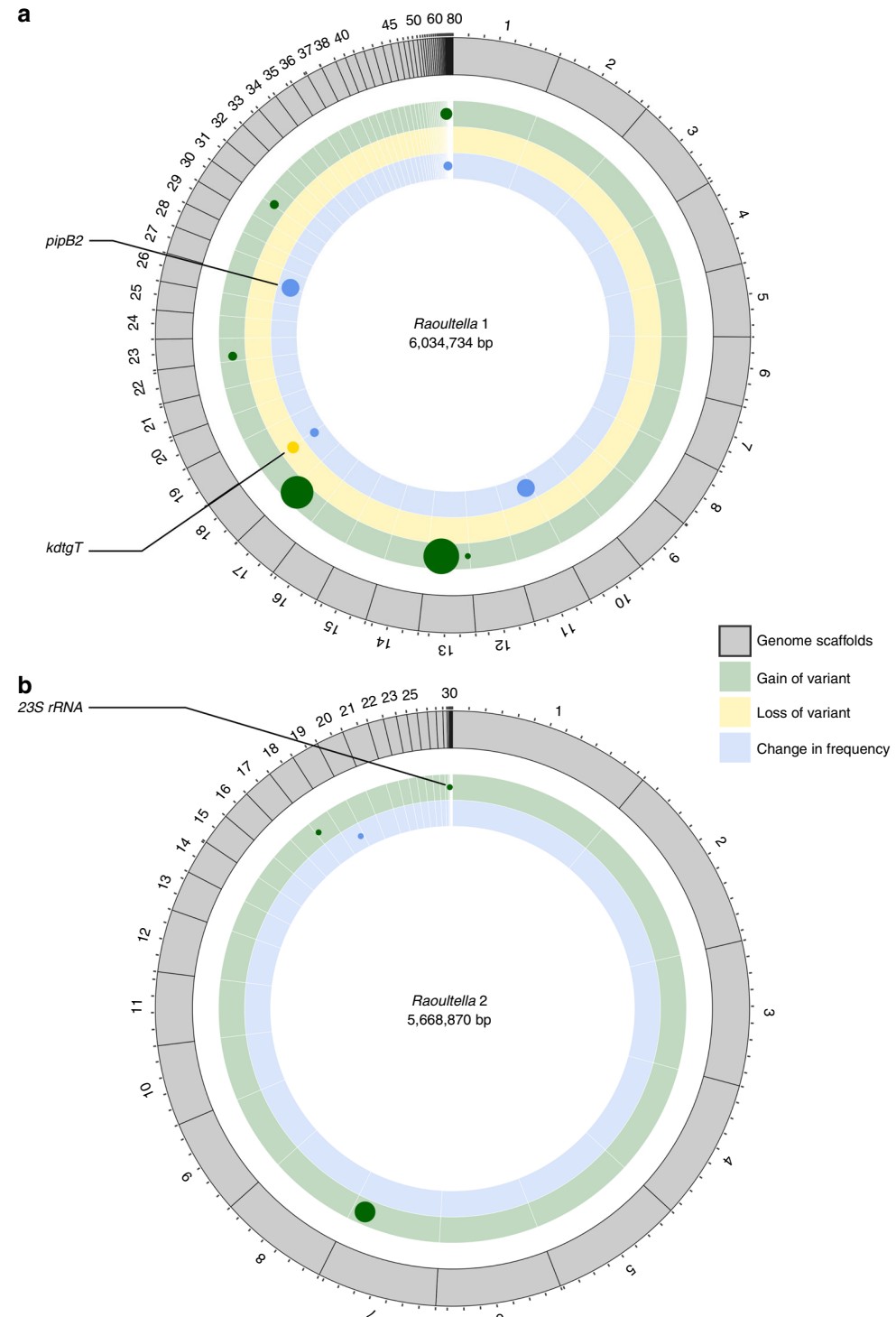

**Fig. 4 Genetic variants found in evolved *Raoultella* focal strains from different background communities. a** Genetic variants detected in *Raoultella* sp.1 focal strains. **b** Genetic variants found in *Raoultella* sp.2 focal strains. Points show significant changes in SNP frequency compared to the ancestral isolate, with colours indicating a significant gain (green), loss (yellow), or significant change in frequency (blue) of genetic variants. Gene names are given for SNPs that lie within known genes. Variants without annotation are hypothetical proteins. The size of the points is proportional to the number of evolved focal strains from the different communities showing the variant class. Source data are provided as a Source Data file.

In this scenario, extrinsic factors provide the opportunity for adaptation, but focal strains can only capitalise on the opportunities if they have the capacity. The interaction between ecological opportunity and evolutionary capacity is likely to have a great influence on how populations will respond to changing

environmental conditions, for example due to global change, or exposure to antibiotics. Our experiment provides a preliminary insight into how bacterial population navigate adaptive landscapes when they are embedded within complex communities, revealing that understanding the biotic environment is vital for

understanding evolutionary trajectories. More work will be needed to disentangle mechanisms of the changes we observed, but future work on evolving communities will inevitably require the joint consideration of intrinsic species variables and external community variables.

## Methods

**Microbial materials.** We sampled water-filled tree-holes between August 2013 and April 2014 from locations across the south of England and selected 11 communities aiming for different levels of biodiversity and robustness (see below and Supplementary Methods for more details). The communities were characterised using 16S amplicon sequencing (Illumina MiSeq, 250 bp-paired end) using primers 515f/806r with the forward primer barcoded, and 97% similarity threshold to count species as Operational Taxonomic Units. We isolated bacterial strains from the communities by plating on R2A agar and identified (Sanger sequencing of 16S locus) individual strains. We selected 22 focal strains (2 per community) that displayed a range of intrinsic properties (see below) and stored these 'ancestral' focal strains at −80 °C.

**Evolution experiment and evolved/ancestor competition.** Standardised densities of the focal strains were caged in dialysis bags, which allow diffusion of molecules but prevent migration of cells into or out of the bags. We also included a negative control and a bag containing a mix of all the focal strains. Each focal strain was suspended in 9 communities (we started with 12 communities but excluded three indicating contamination): 8 complex communities plus a community of the corresponding focal strains itself in the background (see Supplementary Methods for more details). All 216 strain–community combinations were 4× replicated resulting in 864 microcosms. The focal strains were maintained for 5 months. During the experiment, we replaced 10% of the culture medium once a week, which consisted of a beech leaf medium buffered to pH 5.5. At the end of the experiment, we estimated the performance of the evolved focal strains compared to ancestral strains in the presence of the 'co-evolved' communities for a period of 2 weeks, measured as average population increase measuring cell counts by flow cytometry (2592 observations over time, at time 0, 7, and 14 days). We initiated these evolved/ancestor competition experiments with a 1:1 ratio and tracked the ratio change over time. A ratio >1 indicated the evolved strain established higher cell densities than the corresponding ancestor while competing with the community (indicated by a positive slope of the ratio over time of assay comparison).

**Statistical methods to analyse performance data.** Linear-Mixed-Effects models were used to explain these ratio changes with 'strain–community' combination and 'time' as explanatory variables accounting for the pseudo-replication over time with individual microcosms treated as random effects following recipes given elsewhere[50]. Repeated measures ANOVA with focal strain x community as explanatory variables were used to assess the main effects (Pie inset Fig. 1). Detailed methodology and statistical details are in the Supplementary Methods.

**Extrinsic and intrinsic properties and resource usage.** We quantified the extrinsic properties of the communities by quantifying their biodiversity (Shannon's Index) based on 16S rRNA locus sequencing data (see Supplementary Methods for more details). We also measured the ecological robustness of the ancestral community by measuring the difference in instantaneous activity between pH 7 medium and pH 5.5 medium using the Bactiter-Glo assay (Promega). We assessed how feeding niches evolved by measuring the activity of the evolved and ancestral strains for enzymes that metabolise recalcitrant (cellulose), labile (xylose), and intermediate (chitin) substrates using enzyme assay kits as in ref. [38]. For intrinsic properties of the focal strains we measured maladaptation of the ancestral monoculture as the difference in instantaneous activity between pH 7 and pH 5.5 media, as for ecological robustness for communities. We estimated genome size by whole genome sequencing of ancestral clones, quantified mean frequency of the focal strain in the background communities from 16S data, and estimated phylogenetic distance of each focal strain from *Raoultella* sp.2, which we classified as highly evolvable. Detailed methodology and statistical details are in the Supplementary Methods.

**Genome sequencing and changes in community composition.** Evolved and ancestral populations (100 colonies pooled from each focal population to ensure we also detect low changes in frequency) were whole-genome sequenced. We sequenced at least three replicates of *Raoultella* sp.1 and sp.2 across several communities (47 genomes overall) to detect genetic variants (30× coverage, Illumina Hiseq 250 bp paired end sequencing, see Supplementary Methods for details). Reads were mapped onto an assembly of the ancestral genomes to identify single nucleotide polymorphisms (SNPs) both among the 100 pooled colonies and between ancestor and evolved isolates. To assess changes in composition of the background communities, we re-sequenced 16S rRNA locus for several of the final communities. For each of the 8 background communities, we selected the first two replicates containing dialysis bags for control (empty bag), the mix of focal strains, *Rizobium*.1 and *Novosphingobium*.2 resulting in data for 64 background

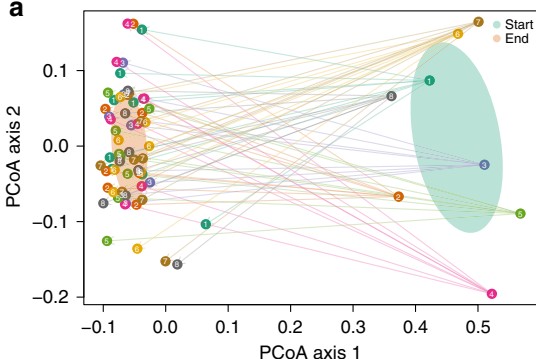

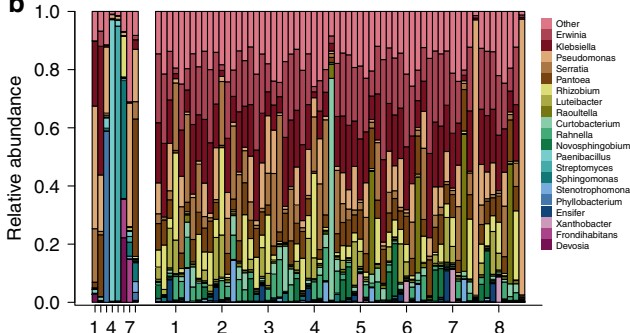

**Fig. 5 Community composition change from start to end of the selection experiment. a** Ordination of the communities using principal coordinates analysis to investigate change in community composition from the start to the end of the selection experiment. Numbers and arrows indicate the communities. Note that we sequenced two replicates from all background communities for four different focal strain bags. **b** Histogram of the most common OTUs highlighting the most important taxa in the final communities. Rarer taxa are subsumed into the "other" category. 'Start' shows the initial communities at month 0 and 'End' the evolved communities after 5 months. Those 'End' communities were used to compete the evolved and ancestral strains. Source data are provided as a Source Data file.

communities. We used principal coordinates ordination to visualise differences in the communities.

The sequencing data (Fig. 5) indicated 1 synonymous substitution across 47 populations. Assuming a mutation rate of $10^{-10}$ per nucleotide per generation[51], and that synonymous substitutions are neutral, we estimated roughly 106 generations had elapsed over the course of the experiment. This estimate relies on several assumptions and therefore is considered to be preliminary, but it provides a rough estimate of generation numbers for comparison with other studies.

**Reporting summary.** Further information on research design is available in the Nature Research Reporting Summary linked to this article.

## Data availability

All relevant data are available from the corresponding author upon request. Community sequencing data that support the findings of this study have been deposited in the NCBI Short Read Archive (project number PRJNA453972), with further information available in ref. [44]. Genome sequences and annotations have been deposited at ENA under the Project accession ID PRJEB34793. Source data underlying the results shown in all the figures of this manuscript are available in the Source Data file.

## Code availability

The codes used for analysing data are available from the corresponding author upon request.

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

## Acknowledgements

The research was supported by the Natural Environment Research Council (NERC, No. NE/K006215/1) of the UK to T.B. T.B. was supported by a Royal Society University Research Fellowship. T.S. finalised the manuscript during his paternity leave supported by the ZBFS. Genome sequencing was provided by MicrobesNG (http://www.microbesng.uk), which is supported by the BBSRC (Grant no. BB/L024209/1).

## Author contributions

T.S., T.B., and T.G.B. designed the study, D.W.R. and T.S. sampled communities and isolates, T.S. and M.H. performed the experiments, T.S. and T.B. analysed the experimental data, R.W.N., T.G.B., and T.S. analysed the genomes, D.W.R., T.B., and T.S. analysed the community sequencing data, T.S., T.G.B., and T.B. wrote the manuscript, with input from all other authors.

## Competing interests

The authors declare no competing interests.
