## [Peer Review File · Nature Communications]

Reviewers' Comments:

Reviewer #1:

Remarks to the Author:

This is a very interesting ms addressing a very important question in (microbial) evolution: how species evolve within communities is unknown, and this is the most substantial experimental study of this to date. The ms uses a very innovative experimental design, embedding a range of natural isolates in a community using dialysis bags allowing their recovery. Comparison of a large number of focal species within a range of community structures enables tests of the intrinsic (ie species-level) and extrinsic (ie environment/community) drivers of evolutionary potential. This represents a huge volume of work which is distilled into a readable and reasonably clear ms. The ms is well-written and the analysis on the whole appears valid and reasonable. The findings are understandably complex although some simple rules do emerge, and although some of the proxies used for e.g. species traits have their limitations, I am firmly of the view that these should be overlooked as this work is truly ground-breaking.

My critical comments are fairly limited and should be easily addressed in revision, they are listed in the order that they appear in the ms:

line 58 - define what you mean by "preiodically"

line 66 - why is performance in inverted commas? You measured competitive fitness, so why not just say so?

line 85 - robustness is little mentioned in the following text; please state the relationship with robustness more obviously/clearly

line 91/2 - This a rather vague statement; clarify what you mean, what is the pattern?

line 126/7 - The causality is unknown here I think? Would it be more reasonable to simply state that there positive/negative associations between the variables?

line 154/7 - genes names should be italicised; also please briefly state what the genes do and state more specifically the functions they are predicted to be involved in

line 182 - "populations"

Figure 1 - this is a horrible figure, the overlap of blocks makes it hard to read; please use another way (colour intensity? darkness?) to signify the significance

Figure 4a - I find the before and after ellipses fairly uninformative, what I want to know is the relationship between starting and end communities, can you show these vectors using arrows? Did you perform e.g. a permutational manova to test for differences in community structure at the end?

Table 1 - this is a fairly uninformative table; at least put the known gene names on the table. I think this information could be better shown plotted on the circular chromosome, but I guess this may not be possible if an assembled genome is unavailable?

References - cite/discuss Hall et al Evolution Letters 2018 doi.org/10.1002/evl3.83 which is relevant to your study

Reviewer #2:

Remarks to the Author:

In this manuscript Scheuer et al perform a series of laboratory microcosmos experiments to gain insights on how communities composed of different bacterial species affect the evolutionary dynamics of a focal strain. This is an important issue, as there are many more studies where bacterial evolution under specific abiotic stresses is followed in real time in the absence of other species, than under the more natural condition of evolution within an ecosystem. To do this, the authors use several collections of strains, which were sampled from a natural ecosystem and that can be passaged in the lab in a semi-natural medium.

The authors show that the increase in performance of a focal strain depends on: the diversity of the ecosystem in which the strain is embedded, its genome size and initial fitness, and the interaction between these intrinsic properties of the strain and the community composition.

The novelty and strength that this study brings is the simultaneous consideration of multiple communities and several focal strains in the same abiotic environment.

The weakness is that the study is mostly correlative, albeit being experimental and thus possible to address results at a deeper level. Indeed very few attempts to gain mechanistic explanations for the observations are made. This absence of attempt at mechanism and the fact that the particular ecosystems are difficult to reproduce experimentally (as the sampling was random and lacks control of the number of initial species) somehow weakens the generality of the conclusions and broadness of interest.

Major comments:

1) The experimental design presented (on FigS1) is unclear on the source of the ancestral focal strains. If I understood correctly there are 11 sampled communities (from nature at different time points) and 22 focal strains from these communities. Not clear is if these are 2 strains per community and what was the rationale on the sampling chosen.

2) The relevance of the environmental pressure applied here are not explained. The dependence of the results on that pressure should be discussed.

3) What is the ballpark estimate for the number of generations that have passed during the 5 months? This is important to assess evolutionary time.

4) It was not clear if the authors could close the genomes of the ancestral strains with their sequencing approach, and if mutations or horizontal transfer events may have been missed.

5) Also it was not clear how the initial frequency of the focal strain influences its evolutionary pattern.

6) Line 87-89: low diversity was associated with increased phenotypic evolution". Was it also associated with increased genotypic evolution?

7) Lines 95-99: I believe the authors have the capacity for experimentally testing at least one of these mechanisms (competition or drift), and thus provide the readers with a more in depth understanding (e.g. by supplementing the media with some resource).

Minor comments:

Table 1 is missing a legend.

Also in Table 1, are the authors sure that the data collection allows to measure changes in frequency as low as 5%?

Figure 2: definitions of shannons, shannons2, phylogne, and phylogen2 are missing

Reviewers' comments: **Our responses are highlighted in bold**

Reviewer #1 (Remarks to the Author):

This is a very interesting ms addressing a very important question in (microbial) evolution: how species evolve within communities is unknown, and this is the most substantial experimental study of this to date. The ms uses a very innovative experimental design, embedding a range of natural isolates in a community using dialysis bags allowing their recovery. Comparison of a large number of focal sepcies

within a range of community structures enables tests of the intrinsic (ie species-level) and extrinsic (ie environment/community) drivers of evolutionary potential. This represents a huge volume of work which is distilled into a readable and reasonably clear ms. The ms is well-written and the analysis on the whole appears valid and reasonable. The findings are understandably complex although some simple rules do emerge, and although some of the proxies used for e.g. species traits have their limitations, I am firmly of the view that these should be overlooked as this work is truly ground-breaking.

Response: We were pleased to receive such positive feedback.

My critical comments are fairly limited and should be easily addressed in revision, they are listed in the order that they appear in the ms:

line 58 - define what you mean by "preiodically"

Response: We replaced 10% of the culture medium once a week to simulate natural conditions. In most experimental evolution studies, a small percentage of the population is transferred to fresh media to maintain high growth rates. However, this procedure results in a substantial population bottlenecks and does not mimic conditions in most natural ecosystems. We now clearly state our replacement procedure in the text (L. 60-62, Supp Methods L97-99).

line 66 - why is performance in inverted commas? You measured competitive fitness, so why not just say so?

Response: We clarify our usage of performance as suggested by the reviewer (L. 71-73).

line 85 - robustness is little mentioned in the following text; please state the relationship with robustness more obviously/clearly

Response: Our analysis indicated a link between robustness and resource usage (evolved strains had a greater capacity to use chitin when they were in communities with low robustness), which is now mentioned in the main text (L. 104-108).

line 91/2 - This a rather vague statement; clarify what you mean, what is the pattern?

Response: We now clarify that there is a tendency for evolution of focal strains to increase at low levels of biodiversity (L. 93-95).

line 126/7 - The causality is unknown here I think? Would it be more reasonable to simply state that there positive/negative associations between the variables?

Response: We have chosen our wording more carefully (L. 127-130) and have used "association" where appropriate as suggested by the reviewer.

line 154/7 - genes names should be italicised; also please briefly state what the genes do and state more specifically the functions they are predicted to be involved in

Response: Gene names have been italicised. We now also briefly state predicted functions (L. 163-166), though most functions are hypothetical for these non-model isolates.

line 182 - "populations"

Response: Agreed, changed as requested (L. 197).

Figure 1 - this is a horrible figure, the overlap of blocks makes it hard to read; please use another way (colour intensity? darkness?) to signify the significance

Response: The figure has been re-configured as suggested by the referee. We now use colour intensity to indicate significance as suggested.

Figure 4a - I find the before and after ellipses fairly uninformative, what I want to know is the relationship between starting and end communities, can you show these vectors using arrows? Did you perform e.g. a permutational manova to test for differences in community structure at the end?

Response: We now include arrows between "start" and "end" communities as suggested. We note that the community numbers are indicated in the plotting symbols. We have conducted a PERMANOVA analysis as suggested and find there is no overall effect of community identity on final community structure, which we note in the text (L184-185).

Table 1 - this is a fairly uninformative table; at least put the known gene names on the table. I think this information could be better shown plotted on the circular chromosome, but I guess this may not be possible if an assembled genome is unavailable?

Response: As suggested by the reviewer we replaced the table with an additional figure plotting circular genomes (Figure 4). The table provides some complementary information and so has been moved to the Supplementary Information.

References - cite/discuss Hall et al Evolution Letters 2018 doi.org/10.1002/evl3.83 which is relevant to your study

Response: We are grateful for the referee pointing out this relevant reference, which has now been added and is now mentioned in the main text.

Reviewer #2 (Remarks to the Author):

In this manuscript Scheuer et al perform a series of laboratory microcosmos experiments to gain insights on how communities composed of different bacterial species affect the evolutionary dynamics of a focal strain. This is an important issue, as there are many more studies where bacterial evolution under specific abiotic stresses is followed in real time in the absence of other species, than under the more natural condition of evolution within an ecosystem. To do this, the authors use several collections of strains, which were sampled from a natural ecosystem and that can be passaged in the lab in a semi-natural medium.

The authors show that the increase in performance of a focal strain depends on: the diversity of the ecosystem in which the strain is embedded, its genome size and initial fitness, and the interaction between these intrinsic properties of the strain and the community composition. The novelty and strength that this study brings is the simultaneous consideration of multiple communities and several focal strains in the same abiotic environment. The weakness is that the study is mostly correlative, albeit being experimental and thus possible to address results at a deeper level. Indeed very few attempts to gain mechanistic explanations for the observations are made.

We appreciate the reviewer comments and are pleased that they find novelty in the work. We agree that there is much more work to be done on mechanisms. In this study, we opted to identify the major factors that contributed to evolution of the focal strains and we will indeed look at mechanisms in future work. We do believe we have marshalled a substantial body of work that is consistent with particular mechanisms (e.g. resource competition) and will serve to narrow down further experiments.

This absence of attempt at mechanism and the fact that the particular ecosystems are difficult to reproduce experimentally (as the sampling was random and lacks control of the number of initial species) somehow weakens the generality of the conclusions and broadness of interest.

The ecosystem we used is very simple to reproduce by other researchers by boiling a few beech leaves, as detailed in the Methods. We believe using non-model organisms collected from nature broadens interest because the results should apply to any set of bacteria and bacterial communities collected from nature in a similar manner – but if a researcher so wished they could repeat it exactly using our frozen focal strains and communities. The goal was not to control the number of initial species but to use the natural variation in species number. Natural communities contain thousands of species, and much of the diversity cannot be cultured/isolated, making it unfeasible to control the initial number of species while retaining natural levels of diversity. There is therefore a tradeoff between using more natural levels of diversity, or precisely controlling the number of species. For this study, we chose the first option, which has been much less well studied.

Many previous studies have controlled the number of starting species as suggested by the reviewer (including several by the authors), but always with fewer species than in real bacterial communities. The novelty and advance of our work (as recognised by both referees) is that we were able to compare the evolution of focal species embedded within communities with natural levels of diversity. We are grateful for the observations and have tried to make these points more clearly in the text.

Major comments:

1)The experimental design presented (on FigS1) is unclear on the source of the ancestral focal strains. If I understood correctly there are 11 sampled communities (from nature at different time points) and 22 focal strains from these communities. Not clear is if these are 2 strains per community and what was the rationale on the sampling chosen.

Response: The reviewer is correct: we sampled 11 communities from nature and isolated 2 strains per community (i.e. 22 strains in total). In terms of the rationale for choosing these strains:

We spread the strains across the communities to increase the generality of the results, and to avoid the bias that would be generated by selecting strains from a limited number of communities, which might turn out to have fast- or slow-evolving species, so that we could test whether "community of origin" had a significant impact on evolvability.

We were interested in understanding the impact of intrinsic and extrinsic factors across the range of values found in nature. The specific strains used were therefore selected to have a range of genome sizes, degrees of maladaptation, abundance, etc.

We have clarified these aspects of the experimental design (L. 213-216).

We have also added further explanations to the Supplementary Information.

2) *The relevance of the environmental pressure applied here are not explained. The dependence of the results on that pressure should be discussed.*

Response: pH is among the most important environmental factors in microbial communities and is the major driver of microbial community structure across many ecosystems. In addition, the environment from which the communities were taken varies substantially in pH, so pH is likely to be an important selective pressure. We have added this clarification to the text (Line 55-62).

Although different selective pressures would undoubtedly produce different results in terms of which specific strains/genes evolved under the different treatments, our prediction is that the general conclusions would hold- i.e. that most of the variation in evolvability would be explained by the Strain x Community interaction, and by a combination of extrinsic and intrinsic factors. We highlight this point in the text. We have clarified this point in the Discussion (Line 187-188).

3) *What is the ballpark estimate for the number of generations than have passed during the 5 months? This is important to assess evolutionary time.*

Response: We agree with the reviewer that it is interesting to estimate the number of generations. However, assessing the number of generations is difficult in our experiment (and in nature) because most of the cell division occur during stationary phase, meaning that we cannot assess generations simply by counting differences in cell numbers over time (the method that is usually used in studies of experimental evolution). However, we can get a ballpark estimate from the molecular data if we make some simplifying assumptions.

We assume that the per-nucleotide mutation rate is comparable to model bacteria:

mutation rate = 10^{-10} mutations/nucleotide/generation

We inspected only neutral sites, which we defined as: loci that were in non-coding regions, and nucleotides that were in the 3rd codon position. This latter assumption was for simplicity of calculation and would actually be somewhat lower, making our estimate conservative. The *Raoultella* genomes were $\sim 6 \times 10^6$ nucleotides, which yielded:

$\sim 2 \times 10^6$ nucleotides at synonymous sites

We assumed that any SNP that was at 0 frequency at the start and at 1.0 frequency at the end is a substitution. For neutral mutations, the substitution rate per locus should equal the mutation rate (cf. Kimura's neutral theory). The number of generations should then equal:

number of substitutions / (number of synonymous sites X 47 replicate populations X mutation rate)

In *Raoultella* 1 in Figure 4 and Table S4, there are no cases of a final synonymous site with frequency 1 starting with frequency 0. In *Raoultella*2 there are 2, but one of them is parallel in multiple replicates, which is highly unlikely to be truly neutral. We believe this site is likely linked to a selected site or is having a selective effect.

There is therefore 1 substitution across all the populations, yielding:

$1/(2 \times 10^6 * 47 * 10^{-10}) = 106$ generations.

The experiment was over 5 months (~ 150 days), which equates to a generation every ~ 36 hours.

This number fits with our intuition about the metabolic activity within the microcosms, with growth rates well below the high growth rates observed for bacteria growing in exponential phase in rich broth (~3 hours per generation), but well above the minimum possible number of generation (which would assume re-growth following sampling but zero generations at carrying capacity).

We recognise a great many assumptions go into this calculation, so we did not want to emphasise this in the text. We provide a brief summary of this estimate in the Methods (Line 273-277).

4) It was not clear if the authors could close the genomes of the ancestral strains with their sequencing approach, and if mutations or horizontal transfer events may have been missed.

Response: We were unable to close ancestral genomes entirely and now mention this in the text (Supplementary Information L. 278-280). The pore size of the dialysis bags prevents horizontal gene transfer, which we now note in the main text (Line 51). We sequenced a pooled collection of 100 isolates per sample and used minimum 30x fold coverage. Genetic variants that were rare (<1/100) would have been unlikely to have been detected, but variants experiencing positive selection would have had ample time to spread assuming our ballpark estimate of 106 generations is reasonable.

5) Also it was not clear how the initial frequency of the focal strain influences its evolutionary pattern.

Response: The initial frequency of the focal strain was standardised across the microcosms, so this should not have influenced its evolutionary pattern. We clarify this in methods (Line 223).

6) Line 87-89: low diversity was associated with increased phenotypic evolution". Was it also associated with increased genotypic evolution?

Response: There was no obvious pattern from the 2 *Raoultella* strains that we sequenced (Table 1), though we believe it is too early to rule out this connection. We explain potential explanations in the main text (Line 170-178).

7) Lines 95-99: I believe the authors have the capacity for experimentally testing at least one of these mechanisms (competition or drift), and thus provide the readers with a more in depth understanding (e.g. by supplementing the media with some resource).

Response: We support the referee's idea, but we believe further experimental tests would be outside of the scope of what is already an enormous experimental effort. In addition, separating drift and competition is complex in these systems: for example, while supplementing the media might alleviate resource competition, it would also select for different communities thus altering the selection experienced by the focal strains. In addition, supplementing the media would not remove other forms of competition (e.g. interference competition), thus selection could still be operating even when resources are not limiting. We have added the reviewer suggestion to the text as the next step for this kind of research (Line 127-131).

Minor comments:

Table 1 is missing a legend.

Response: Apologies, the table caption was omitted. We now add the information in form of figure 4.

Also in Table 1, are the authors sure that the data collection allows to measure changes in frequency as low as 5%?

Based on sampling 100 pooled isolates in each sequencing run with 30x coverage, we could estimate frequencies that were as low as 1-3.3%.

Figure 2: definitions of shannons, shannons2, phylogne, and phylogen2 are missing

Response: The Shannon Index and phylogenetic distance are now defined in the Supplementary Methods. Shannons² and Phylogeny² are simply the square of these values. (Supporting Information L. 173-188).

Reviewers' Comments:

Reviewer #1:

Remarks to the Author:

The authors have done a thorough job of revising their ms and all my comments have been fully and satisfactorily addressed. I have no further comments.

Reviewer #2:

Remarks to the Author:

The authors have made substantial clarifications in the new version, which have resolved the issues I had previously raised.

I thank the authors for this effort and I believe that with this added explanations, the design used in this paper can now be better appreciated by the broader community.

I also appreciated very much the ballpark estimate of the number of generations in the experiment. As for mechanisms, I appreciated the new directions now mentioned in the manuscript towards these, and fully understand that it will be future work which requires a considerable experimental effort.

Two minor points that the author may wish to do:

line 138 evolution to use to xylose -> evolution to use xylose

line 146 after "insufficient time for significant evolutionary change" add (see Methods).

Response Letter Reviewers Requests NCOMMS-19-13445A

We are delighted our manuscript entitled "Bacterial adaptation is constrained in complex communities" has been positively reviewed. We revise the manuscript, the figures, the supporting information and hope we were able to address all comments from the editor and the reviewers. Please find our detailed responses below in **bold**.

REVIEWERS' COMMENTS:

Reviewer #1 (Remarks to the Author):

The authors have done a thorough job of revising their ms and all my comments have been fully and satisfactorily addressed. I have no further comments.

RE: We are pleased to receive such positive feedback.

Reviewer #2 (Remarks to the Author):

The authors have made substantial clarifications in the new version, which have resolved the issues I had previously raised.

I thank the authors for this effort and I believe that with this added explanations, the design used in this paper can now be better appreciated by the broader community.

I also appreciated very much the ballpark estimate of the number of generations in the experiment.

As for mechanisms, I appreciated the new directions now mentioned in the manuscript towards these, and fully understand that it will be future work which requires a considerable experimental effort.

RE: We are pleased to receive such positive feedback.

Two minor points that the author may wish to do:

line 138 evolution to use to xylose -> evolution to use xylose

RE: Changed accordingly (p.6 L.154)

line 146 after "insufficient time for significant evolutionary change" add (see Methods).

RE: Added the phrase (p.6 L.163)